# Effects of distancing and pattern of breathing on the filtering capability of commercial and custom-made facial masks: An in-vitro study

**Lorenzo Ball**[1], **Stefano Alberti**[2]*, **Claudio Belfortini**[2], **Chiara Almondo**[1], **Chiara Robba**[1], **Denise Battaglini**[1], **Carlo Cravero**[3], **Paolo Pelosi**[1], **Valentina Caratto**[2], **Maurizio Ferretti**[2]

**1** Department of Surgical Sciences and Integrated Diagnostics, University of Genoa, Genova, Italy,
**2** Department of Chemistry and Industrial Chemistry, University of Genoa, Genova, Italy, **3** Department of Mechanical, Energetic, Management and Transports Engineering, University of Genoa, Genova, Italy

\* stefano.alberti@edu.unige.it

## Abstract

### Background

Since the beginning of the COVID-19 pandemics, masking policies have been advocated. While masks are known to prevent transmission towards other individuals, it is unclear if different types of facial masks can protect the user from inhalation. The present study compares in-vitro different commercial and custom-made facial masks at different distances and breathing patterns.

### Methods

Masks were placed on a head mannequin connected to a lung simulator, using a collecting filter placed after the mannequin airway. Certified, commercial and custom-made masks were tested at three different distances between the emitter and the mannequin: 40 cm, 80 cm and 120 cm. Two patterns of breathing were used, simulating normal and polypneic respiration. A solution of methylene blue was nebulized with a jet nebulizer and different mask-distance-breathing pattern combinations were tested. The primary endpoint was the inhaled fraction, defined as the amount of methylene blue detected with spectrophotometry expressed as percent of the amount detected in a reference condition of zero distance and no mask.

### Findings

We observed a significant effect of distance ($p < 0.001$), pattern of breathing ($p = 0.040$) and type of mask ($p < 0.001$) on inhaled fraction. All masks resulted in lower inhaled fraction compared to breathing without mask ($p < 0.001$ in all comparisons), ranging from 41.1% ± 0.3% obtained with a cotton mask at 40 cm distance with polypneic pattern to <1% for certified FFP3 and the combination of FFP2 + surgical mask at all distances and both breathing pattern conditions.

**Data Availability Statement:** All relevant data are within the paper and its Supporting Information files.

**Funding:** The authors received no specific funding for this work.

**Competing interests:** The authors have declared that no competing interests exist.

## Discussion

Distance, type of device and breathing pattern resulted in highly variable inhaled fraction. While the use of all types of masks resulted relevantly less inhalation compared to distancing alone, only high-grade certified devices (FFP3 and the combination of FFP2 + surgical mask) ensured negligible inhaled fraction in all conditions.

## Introduction

Since December 2019, the pandemic of coronavirus disease (COVID-19), sustained by the severe acute respiratory syndrome coronavirus-2 (SARS-CoV-2) had a dramatic impact on healthcare systems as well as people's lifestyle and social behavior [1]. This disease is transmitted through several routes, but the respiratory route is the principal one. Half of patients 2019-nCoV RNA was detected, thus concluding that saliva droplets and water aerosol may actually bring living virus [2]. In fact, the disease is mainly transmitted through the emission of droplets during coughing and sneezing, while the airborne route is more debated [3]. The distinction between these two routes is based on the size of emitted particles, with a cut-off diameter commonly set at 5 μm: droplets which are larger than 5 μm contain a large amount of virus but can spread for a shorter distance, while aerosols of smaller size could remain in the ambient air in suspension resulting in contagious aerosols [3, 4]. Also normal speaking may cause airborne virus transmission in confined environments, where small droplets likely play a major role [5].

Shortage of personal protection equipment (PPE) has heavily affected the healthcare systems in the early phases of the pandemics and have led to the public use of a variety of solutions which are generally of unknown efficiency [6]. Masks reduce the spread and transmission of respiratory particles and droplets potentially containing viruses and their use in the general population has been adopted in most countries, even though still under debate [7, 8]. In general, while the gold standard for personal protection is represented by certified respirators [9], their high cost and low availability brought interest towards cheaper alternatives such as surgical masks as well as custom-made protection devices, especially for non-healthcare workers and general population; despite the characterization of their efficacy could represent a significant knowledge for medical professionals working in areas with a low but still real risk of contamination. The National Institute for Occupational Safety and Health defined N95 and N99 standards in the United States and the European Committee for Standardization specified with the EN 143 regulation standards for type 1, 2 and 3 filtering face pieces (FFP1, FFP2 and FFP3). While there is an undoubted role of facial masks as mean to reduce the spread of aerosols and droplets from unaware infected patients, it is unclear whether non-PPE devices can offer some degree of protection to healthy individuals [10, 11].

We aimed to compare the filtering capabilities of commercial and custom-made facial masks in an in-vitro model of mixed airborne-droplet aerial transmission. We hypothesized that the distance from the emitter and the pattern of breathing could affect the protection offered to a model of spontaneously breathing adult by the different types of investigated devices.

## Methods

### Experimental model

As illustrated in **Fig 1**, the emitter was simulated using a Hudson Micromist (Teleflex, US) small volume jet nebulizer operated at 7 L/m driving flow. Under these conditions, the

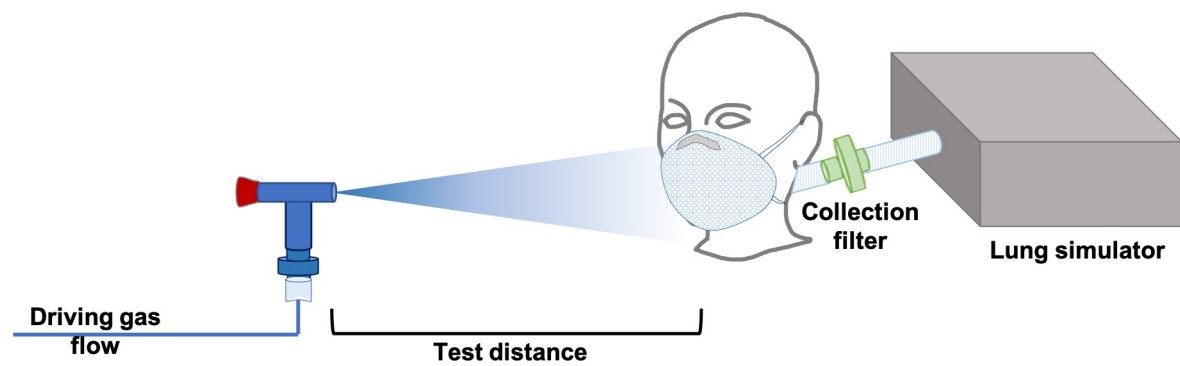

**Fig 1. Experimental setting.**

nebulizer generates a distribution of drops with a median size of 5 μm and an asymmetric distribution [12] thus covering the range of both airborne and droplet transmission. As marker, we nebulized a solution of 3.44 g/L of methylene blue.

The receiver was simulated with a pneumatic lung simulator (Dimar, Mirandola, Italy) connected to a head mannequin to simulate an exposed individual, protected with different masks. We simulated two respiratory patterns: normal breathing with 550 mL tidal volume and respiratory rate of 14 min$^{-1}$, and mildly polypneic pattern with 550 mL tidal volume and respiratory rate of 20 min$^{-1}$.

The emitter was placed in front of the receiver at three distances: 40, 80 and 120 cm. Between the receiver and the lung simulator (Fig 1), a custom-made filter with disposable cotton pads, as described in a previous study [13], was placed to intercept the nebulized methylene blue. The collecting filter was placed after the mannequin airways, to estimate the amount of nebulized particles inhaled by this model of spontaneously breathing subject. The nebulizer was operated for 40 min loaded with 16 ml of methylene blue solution refilling the nebulizer to its maximum capacity during two brief interruptions of nebulization lasting <10 seconds, and each mask-distance-breathing pattern combination was measured three times, in replicate. The experimental time of 40 min was chosen to simulate a scenario of prolonged person-to-person exposure in a closed environment, such as occurs in several social settings. The nebulization time was also titrated in order to achieve sufficient sensitivity for the measurement of inhaled fraction at the precision level of 1%.

As detailed in **Table 1**, we tested certified PPEs, commercially available non-PPE masks and custom-made devices and compared them to the effect of distancing alone, without mask. **Fig 2** shows the tested masks, non-PPEs and custom-made devices in all examined configurations (i.e., alone and their combinations): Surgical mask, FFP1, FFP2, FFP3, FFP2 + Surgical mask, Malpositioned FFP2 (uncovered nose), Cotton mask, Dusting Cloth mask and Cotton + Dusting mask.

### Inhaled fraction measurement

The amount of methylene blue deposed on the cotton pads was analyzed with ultraviolet-visible spectrophotometry (Lambda 35, Perkin Elmer, US), operated at a wavelength of 664 nm, corresponding to the minimum transmittance of methylene blue [14]. We compared the amount detected at each mask-distance-breathing pattern combination with the amount detected with the emitter placed directly in contact with the mannequin airway opening, considered as reference (100% inhaled fraction). A blank cotton pad was used as reference for 0% inhaled fraction. The filtering capability was expressed as the percent inhaled fraction of

**Table 1. Detailed description of the different types of tested devices, according to their certification, commercial name and manufacturer.**

| Short name | Type of device | Description | Commercial name | Manufacturer |
|---|---|---|---|---|
| Surgical mask | Commercial | Multilayer surgical mask | Non-woven face mask | Sinomedic, Serbia |
| FFP1* | Commercial, certified | PPE certified as FFP1 | Shell Mask XMASC | Icoguanti S.p.A., Italy |
| FFP2* | Commercial, certified | PPE certified as FFP2 | Aura 9320+ | 3M, Minnesota, US |
| FFP3* | Commercial, certified | PPE certified as FFP3 | 2737 FFP3 RD | GVS Filter Technology, UK |
| FFP2* + Surgical mask | Combination of two devices | FFP2 mask plus a covering surgical mask | | |
| Malpositioned (uncovered nose and covered mouth) FFP2* | Commercial, certified, malpositioned | FFP2 placed not optimizing fitting and leaving a small space between nose and mask | | |
| Cotton mask | Commercial, not certified | Two-layers cotton plus non-woven tissue mask | | |
| Dusting cloth mask | Custom-made | Two layers of dusting cloth shaped as face mask | Swiffer | Procter & Gamble, US |
| Cotton + dusting mask | Combination of two devices | Cotton mask plus a covering dusting mask | | |

*FFP stands for Filtering Factor Protection of type 1, 2 and 3, according to EN 143 regulations standards, as depicted in Introduction paragraph.

methylene blue compared to zero-distance and no mask with the two patterns of breathing. We assessed the minimum amount of detectable inhaled fraction (*i.e.*, sensitivity) in a calibration experiment in which nebulization time was decreased in 1 s steps. The minimum nebulization time required to detect a non-zero transmittance was 22 s, corresponding to 0.92% of the nebulization time used in our experimental runs. We therefore assumed a sensitivity of 1%; when the measured inhaled fraction was below this threshold, we assumed an inhaled fraction of 1%.

## Statistical analysis

The filtering capability was reported as percent inhaled fraction compared to distancing alone and aggregated as average ± standard deviation. We used a linear model to investigate the effects of distance, device and pattern of breathing on inhaled fraction. We used the Dunnett test for post-hoc multiple comparisons. All analyses were performed with SPSS 25.0 (IBM, Chicago, Illinois). Statistical significance was considered for two–tailed $p < 0.05$.

## Results

Report from **Fig 3** shows the percent inhaled fraction at 40 cm, 80 cm and 120 cm with all the tested devices and the two patterns of breathing and compared to unmasked distancing alone. At the linear model modeling inhaled fraction as function of the following parameters, we observed a significant effect of distance ($p < 0.001$), pattern of breathing ($p = 0.040$) and type of mask ($p < 0.001$) on the inhaled fraction. All masks resulted in an inhaled fraction lower than that without mask ($p < 0.001$ in all pairwise post-hoc comparisons). However, the inhaled fraction was highly variable among devices, with the highest value of 41.1% ± 0.3% obtained with a cotton mask at 40 cm distance with polypneic pattern and only certified FFP3 and the combination of FFP2 + surgical mask ensuring inhaled fraction below 1% at all distance and breathing pattern conditions. At any distance, all devices had higher filtering capability than distancing alone and the magnitude of differences amongst devices was higher at closer distances. The polypneic pattern of breathing increased the inhaled fraction with all

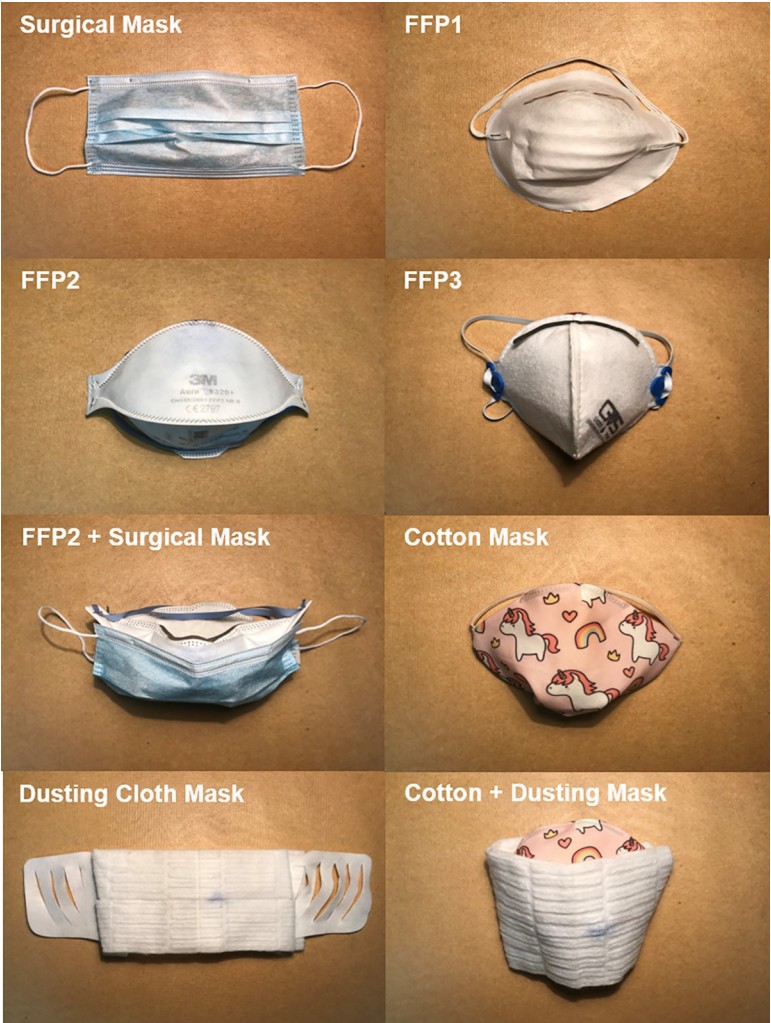

**Fig 2. Tested devices.**

devices except FFP1, FFP2 and FFP3 masks at 40 cm and 80 cm, while at 120 cm we did not observe differences between patterns except for the surgical mask. Also, incorrect positioning of the mask decreased the mask's performances (**Figs 3 and 4**). The dependence of the percent inhaled fraction as a function of the distance and breathing pattern is shown in **Fig 4**.

## Discussion

The main findings of this study are that: 1) all tested masks reduced the inhaled fraction compared to distancing alone, but their efficiency was highly variable; 2) the differences between devices were more pronounced at shorter distances; 3) a polypneic breathing pattern led to an increase in inhaled fraction in most tested conditions.

The determination of masks' efficacy is a complex topic that still represents an active field of research. In the present paper, we developed an inexpensive method to test the filtering capabilities of several face masks, simulating different scenarios and factors affecting their efficiency. In a recent in-vitro study, authors demonstrated that several custom-made masks approached the performances of surgical masks [6]. However, in that study authors focused on

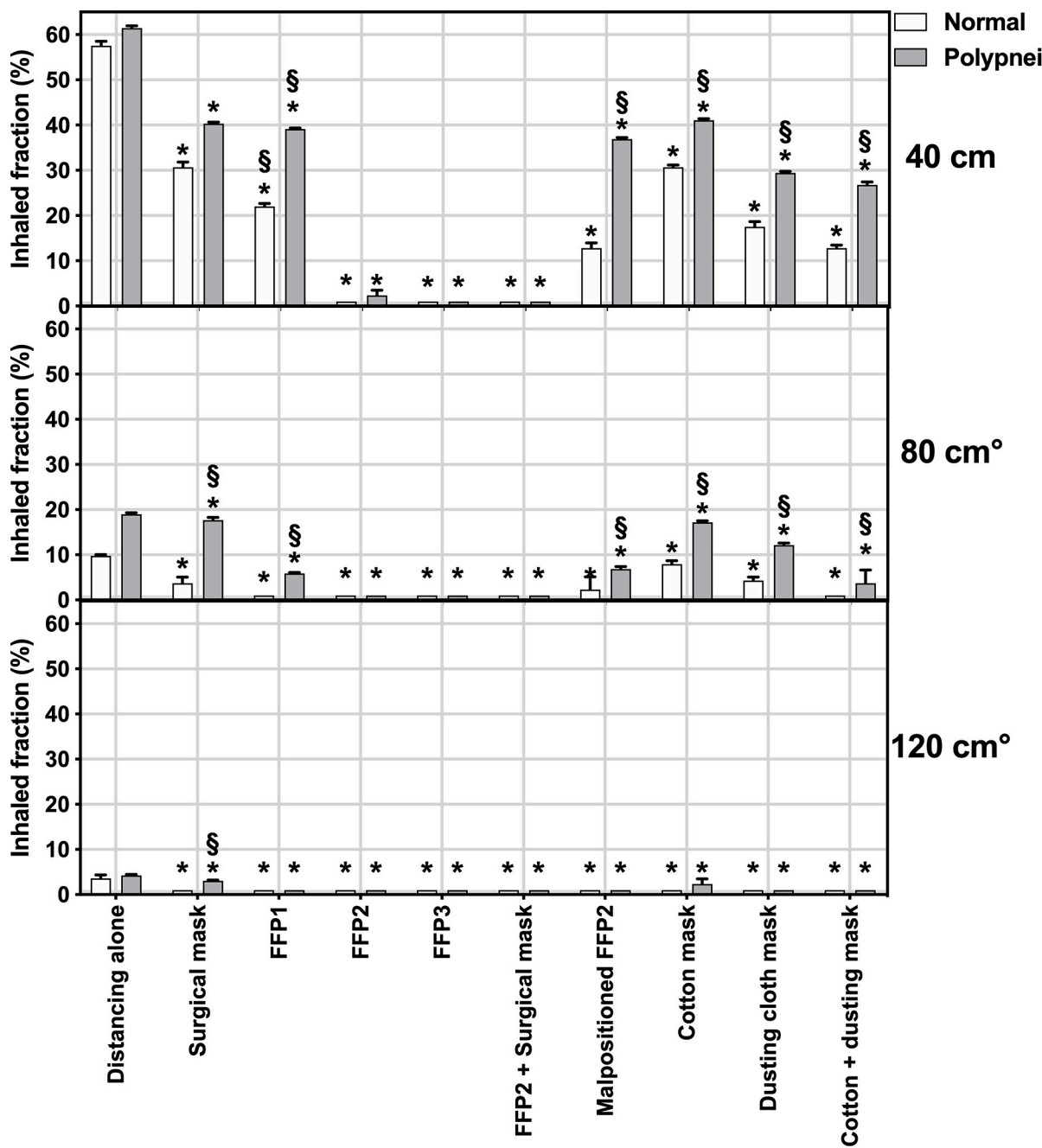

**Fig 3. Effects of different distances and pattern of breathing on the inhaled fraction with the tested masks during normal (white bars) and polypneic (gray bars) pattern of breathing.** The inhaled fraction is expressed as percent of the amount inhaled without mask and at zero distance from the emitter. Bars represent means, error bars the standard deviation. The horizontal gray bar represents the sensitivity limit of our technique. *Significant difference compared to distancing alone (p<0.001), °Significant difference compared to distance of 40 cm, § significant difference compared to normal pattern of breathing at same distance (p<0.05).

the masks' capability of filtering expelled droplets, thus the efficacy of masks in protecting other individuals, while in this study we focused on the inhaled fraction, i.e. the ability of masks to protect the individual from an unknown, unmasked subject.

We investigated the effect of different type of devices considering distance and type of pattern of breathing as factors modifying mask efficacy. The jet nebulizer guaranteed a

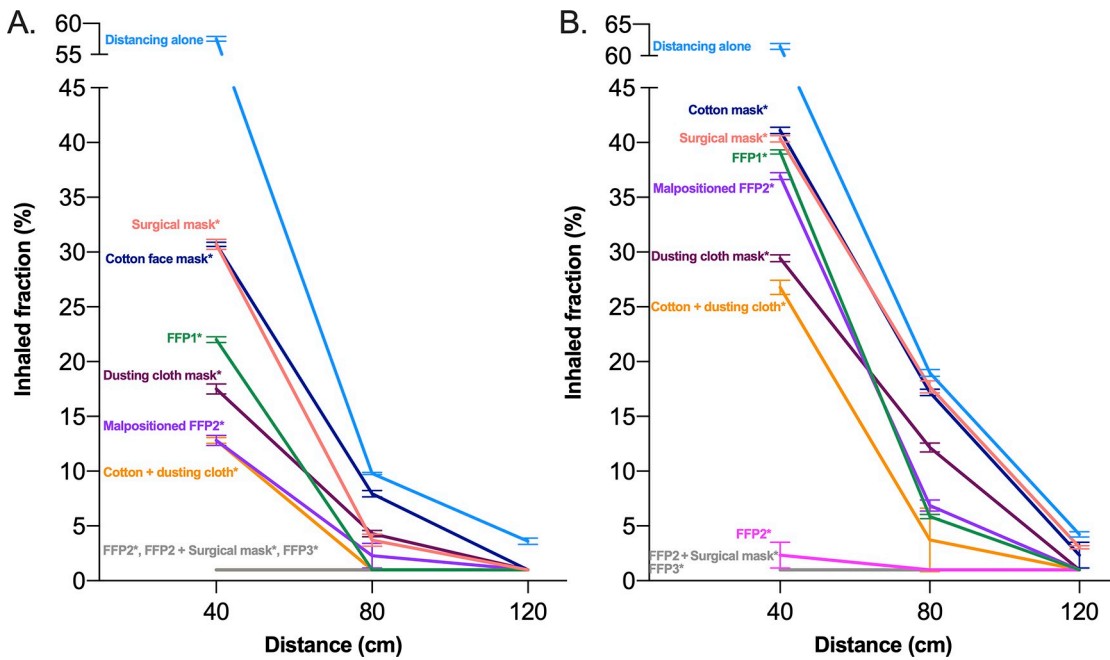

**Fig 4. Effect of distance from the emitter on inhaled fraction with different masks during normal (A) and polypneic (B) pattern of breathing.** *Significant difference compared to distancing alone (p<0.001).

distribution of nebulized particles covering the size range of both droplet and airborne transmission models. While the main route of transmission of COVID-19 seems to be related to larger droplets, smaller drops emitted during normal speaking can survive in suspension in the air, resulting in airborne transmission [15]. Direct airborne transmission due to coughing or sneezing is a complex phenomenon: bigger particles may remain in the surrounding air for several minutes prior to dropping to the ground; while smaller particles tend to remain into the air, thus favoring long-distance diffusion. In case of absent ambient ventilation, these particles will slowly and steadily diffuse throughout the space, remaining in the air for many hours [16]. These mechanisms should be carefully accounted for in this pandemic frame, as they play a crucial role for understanding the airborne transmission of infectious diseases [17, 18].

Considering a short distance as in the 40 cm experimental setting, a typical distance between an infected patient and a healthcare professional or occurring during close social interaction, the only PPE that reduced inhaled fraction to below 1% were only FFP2, FFP3 and FFP2+surgical mask. Moreover, incorrect positioning of the FFP2 mask resulted in a relevant decrease of its filtering capability. Custom made devices based on cotton, dusting cloth and their combinations offered low protection but comparable to that obtained with a surgical or FFP1 mask. Distance alone markedly reduced the inhaled fraction also without masks, thus underlining how physical distancing intrinsically increase the protective effect of all other measures. In fact, at 120 cm from the emitting source, both breathing patterns resulted in an inhaled fraction below 5% also without mask and any type of mask lowered this value below the sensitivity threshold of our technique with a quiet pattern of breathing. On the other hand, at the same distance but with a polypneic pattern, only FFP2, FFP3 and dusting cloth custom-made devices ensured inhaled fraction below 1%. However, also in the worst studied scenario, thus with 40 cm distance and polypneic breathing, also cheap non-certified and custom-made devices reduced the inhaled fraction to 26%-41%, depending on the device, compared to more than 60% obtained without mask. Whether these values are sufficient to actually reduce the

risk of infection is difficult to ascertain, there is rising consensus on the fact that also reduction of initial inhaled viral load could be beneficial in reducing the severity of the disease [19].

This in-vitro study adds to the current knowledge as it explores the hypothesis that, in addition to preventing dispersion of droplets, non-certified devices might anyway offer a low but significant reduction of the inhaled fraction. Previous epidemiologic investigations have suggested a strong relationship between public masking and pandemic control, reducing the growth of the epidemic curve [19]. Our findings suggest that this might also be related to some degree of individual protection, in addition to the known ability of protecting others through reduction of emitted droplets.

This study has limitations that should be addressed. First, it is an in-vitro study, thus we were not able to conclude on actual potential of preventing infections. However, we simulated a distribution of droplets similar to that achieved in the real world, and we tested realistic scenarios and different devices. Second, our technique had a sensitivity of around 1% thus, we cannot conclude on comparisons made between efficient devices, however this sensitivity was sufficient to highlight the differences observed with all the devices commonly used in the general population. Third, we did not investigate the effects of additional factors and devices, such as face shields. However, while of proven efficacy in the healthcare setting [20], their use in the general population is very limited.

## Conclusions

Distance, type of device and breathing pattern affected the protective efficacy of masks. While only high-grade certified devices ensured negligible inhaled fraction in all conditions, the use of all types of masks resulted less inhalation compared to distancing alone.

## Supporting information

**S1 Table. Experimental dataset of percentage inhaled fraction as a function of type of device, breathing pattern and distance.**
(DOCX)

## Author Contributions

**Conceptualization:** Lorenzo Ball, Carlo Cravero, Paolo Pelosi, Valentina Caratto, Maurizio Ferretti.

**Data curation:** Lorenzo Ball, Stefano Alberti, Chiara Robba, Denise Battaglini.

**Formal analysis:** Lorenzo Ball, Paolo Pelosi, Maurizio Ferretti.

**Investigation:** Stefano Alberti, Claudio Belfortini, Chiara Almondo, Valentina Caratto.

**Methodology:** Lorenzo Ball, Carlo Cravero, Paolo Pelosi, Valentina Caratto, Maurizio Ferretti.

**Project administration:** Lorenzo Ball, Carlo Cravero, Paolo Pelosi, Valentina Caratto, Maurizio Ferretti.

**Supervision:** Stefano Alberti, Claudio Belfortini, Chiara Almondo, Valentina Caratto.

**Validation:** Lorenzo Ball, Stefano Alberti, Chiara Robba, Denise Battaglini.

**Writing – original draft:** Lorenzo Ball, Stefano Alberti, Denise Battaglini, Valentina Caratto.

**Writing – review & editing:** Lorenzo Ball, Stefano Alberti, Claudio Belfortini, Chiara Almondo, Chiara Robba, Denise Battaglini, Carlo Cravero, Paolo Pelosi, Valentina Caratto, Maurizio Ferretti.

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
