## [Decision Letter · Decision Letter 0]

8 Mar 2021

PONE-D-20-40104

Effects of distancing and pattern of breathing on the filtering capability of commercial and custom-made facial masks: an “in-vitro” study.

PLOS ONE

Dear Dr. Alberti,

Thank you for submitting your manuscript to PLOS ONE. After careful consideration, we feel that it has merit but does not fully meet PLOS ONE’s publication criteria as it currently stands. Therefore, we invite you to submit a revised version of the manuscript that addresses the points raised during the review process.

We also had some further concerns, which should be addressed in the revised manuscript. PLOS specifies that experiments, statistics, and other analyses are performed to a high technical standard; sample sizes are large enough to produce robust results; and methods are described in sufficient detail to allow another researcher to reproduce the experiment (http://journals.plos.org/plosone/s/criteria-for-publication#loc-3). We are concerned that you have not provided sufficiently detailed methodology for the study to be reproducible. Nor is the rationale clear for some of their methodological choices -- i.e., why do you choose a 40-minute spray time, and is this realistic in terms of real-world scenarios? We also note that you do not specify the amount of time that you pause to refill, nor why you designed the experiment where a refill is necessary.

We are also concerned that the conclusions of your study are overstated, and that they are not directly supported by the results of the experiments. PLOS ONE's fourth publication  criterion (http://journals.plos.org/plosone/s/criteria-for-publication#loc-4) states that the data presented in PLOS ONE manuscripts must support the conclusions drawn. We would like you to clarify that you are not claiming this method reduces infection rates, just the amount of aerosol particulates inhaled. As such, we ask you to revise the language used in your Discussion and Conclusions sections, to ensure that all statements are supported by the results of the study.

We look forward to receiving your revised manuscript.

Kind regards,

Francesco Di Gennaro

Academic Editor

PLOS ONE

Journal Requirements:

Additional Editor Comments:

dear authors follow reviewer suggestions to improve your paper

Reviewers' comments:

Reviewer's Responses to Questions

**Comments to the Author**

1. Is the manuscript technically sound, and do the data support the conclusions?

Reviewer #1: Yes

Reviewer #2: Yes

2. Has the statistical analysis been performed appropriately and rigorously? 

Reviewer #1: Yes

Reviewer #2: Yes

3. Have the authors made all data underlying the findings in their manuscript fully available?

Reviewer #1: Yes

Reviewer #2: Yes

4. Is the manuscript presented in an intelligible fashion and written in standard English?

Reviewer #1: Yes

Reviewer #2: Yes

5. Review Comments to the Author

Reviewer #1: It is a scientifically and methodologically well-articulated study. I found it very relevant to the current global pandemic situation. I recommend acceptance for publication after a minor revision is done. I hope the following clarity concerns will help the authors to improve their manuscript.

Specific comments

1. I suggest ‘physical or social distancing or just simply distance’ to include as a keyword.

2. Page6, 2nd Paragraph

• The nebulizer was operated for 40 minutes and a refill was possible. But it is not clear how long was the exposure time for each test? The test was replicated three times? Was any difference recorded among each test?

• Does exposure time(Length of exposure influenced the percentage of inhalation fraction?

3. Page9, the 1st paragraph of discussion the phrase “ 1) all investigated variables resulted significant (p < 0.001)” need to be clarified. Does it mean all investigated materials resulted in a significant reduction of an inhaled fraction? The result showed that Cotton masks at a 40cm distance with polypneic pattern only reduce to 41.1% ± 0.3%. Can we say this is significant as compared to certified FFP3 or FFP2 + surgical mask which reduce to lower than 1%?

4. Page12, as a limitation, it is mentioned that the sensitivity of the technique for this study is 1%. Can we say this technique is valid or acceptable with this very low sensitivity rate? If I am not mistaken how sensitivity is used in this case, a 1% sensitivity rare makes your methodology unacceptable. Please revised or clarify how the sensitivity of the technique was measured and used in your methodology.

Reviewer #2: First of all it was interesting area of study and I would like to thank the authors. Then I have minor comments regarding your manuscript;

Generally it is better to see your English, some re-arrangements of your reporting and clarity of your statements for every reader.

Please change INTERPRETATION to Discussion in your abstract section.

In result section please start with Report from figure 3 shows…

The pattern of breathing had a negative impact on the filtering capacity: compared to normal breathing... Also, incorrect positioning of the mask decreased the mask’s performances. “Statements that are discussing should be taken to discussion section

°Significant difference compared to distance of 40 cm, § what are the signs at first and end of statements?

“The main findings of this study are that: 1) all investigated variables resulted significant (p < 0.001); 2) all tested masks reduced the inhaled fraction compared to distancing alone; 3) distance decreased the inhaled fraction; 4) the differences between devices were more pronounced at shorter distances; 5) polypneic breathing pattern led to a decrease in most masks’ efficacy; 6) findings clarified the limitations/advantages of using masks both for individual and other protection.” Take this statement to finding section.

[17], [18] your citation should better like this [17, 18].

Some of your statements are not easy to understand for all readers please revise all of your write up…“Distance alone markedly reduced the inhaled fraction also without masks, thus underlining how physical distancing policies intrinsically increase the protective effect of all other measures.”

“Worst experimental condition” what does this mean? Is there such experiment?

6. PLOS authors have the option to publish the peer review history of their article (what does this mean?). If published, this will include your full peer review and any attached files.

Reviewer #1: **Yes: **Serebe Gebrie

Reviewer #2: **Yes: **THOMAS AYALEW ABEBE

---

## [Author Response · Author response to Decision Letter 0]

26 Mar 2021

PONE-D-20-40104

Effects of distancing and pattern of breathing on the filtering capability of commercial and custom-made facial masks: an “in-vitro” study.

PLOS ONE

Dear Dr. Alberti,

Thank you for submitting your manuscript to PLOS ONE. After careful consideration, we feel that it has merit but does not fully meet PLOS ONE’s publication criteria as it currently stands. Therefore, we invite you to submit a revised version of the manuscript that addresses the points raised during the review process.

Reply: We are thankful to the Editor for giving us the opportunity to revise our manuscript. We have modified the manuscript according to the reviewers’ and Editor’s additional comments and we hope you will find the revised manuscript suitable for publication on PLOS ONE.

We also had some further concerns, which should be addressed in the revised manuscript. PLOS specifies that experiments, statistics, and other analyses are performed to a high technical standard; sample sizes are large enough to produce robust results; and methods are described in sufficient detail to allow another researcher to reproduce the experiment (http://journals.plos.org/plosone/s/criteria-for-publication#loc-3). We are concerned that you have not provided sufficiently detailed methodology for the study to be reproducible. Nor is the rationale clear for some of their methodological choices -- i.e., why do you choose a 40-minute spray time, and is this realistic in terms of real-world scenarios? We also note that you do not specify the amount of time that you pause to refill, nor why you designed the experiment where a refill is necessary.

Reply: We agree that part of the methods was unclear and poorly described. We better clarified the experimental setting in the revised version of the manuscript. The choice of 40 minutes of nebulization was to simulate a prolonged person-to-person exposure and also conditioned by the necessity of achieving a sufficient sensitivity to detect inhaled fractions below 1%. The chosen study design required two brief stops of nebulization (<10 seconds) over 40 minutes, we judged these suspensions of nebulization irrelevant to the findings of our study. We hope that these aspects are now better clarified in the revised manuscript. 

We are also concerned that the conclusions of your study are overstated, and that they are not directly supported by the results of the experiments. PLOS ONE's fourth publication criterion (http://journals.plos.org/plosone/s/criteria-for-publication#loc-4) states that the data presented in PLOS ONE manuscripts must support the conclusions drawn. We would like you to clarify that you are not claiming this method reduces infection rates, just the amount of aerosol particulates inhaled. As such, we ask you to revise the language used in your Discussion and Conclusions sections, to ensure that all statements are supported by the results of the study.

Reply: Thank you for pointing this out. We recognize that the original submitted manuscript contained parts that tended to over-interpret our findings. We have extensively revised the abstract and the discussion in order to avoid such over-interpretation.

Reply to Reviewer #1 comments

Reviewer #1: It is a scientifically and methodologically well-articulated study. I found it very relevant to the current global pandemic situation. I recommend acceptance for publication after a minor revision is done. I hope the following clarity concerns will help the authors to improve their manuscript.

 Reply: we thank the reviewer for their positive and constructive comments.

Specific comments

1. I suggest ‘physical or social distancing or just simply distance’ to include as a keyword.

Reply: thank you for your suggestion, we added the suggested keyword.

2. Page6, 2nd Paragraph

• The nebulizer was operated for 40 minutes and a refill was possible. But it is not clear how long was the exposure time for each test? The test was replicated three times? Was any difference recorded among each test?

Reply: Thank you for pointing out this lack of clarity. As also requested by the Editor, we better clarified the experimental design in particular concerning the nebulization time. As better stressed in the revised manuscript, each measurement was performed in triplicate. Reproducibility was good, as highlighted by the standard deviations illustrated in the figures. 

• Does exposure time(Length of exposure influenced the percentage of inhalation fraction?

Reply: as now better clarified in the revised manuscript, the exposure time was constantly kept at 40 minutes, therefore no effect of time could be investigated.

3. Page9, the 1st paragraph of discussion the phrase “ 1) all investigated variables resulted significant (p < 0.001)” need to be clarified. Does it mean all investigated materials resulted in a significant reduction of an inhaled fraction? The result showed that Cotton masks at a 40cm distance with polypneic pattern only reduce to 41.1% ± 0.3%. Can we say this is significant as compared to certified FFP3 or FFP2 + surgical mask which reduce to lower than 1%?

Reply. We recognize that the reporting of statistic was unclear. We now better clarify that the first p-values refer to the linear model using device, distance and pattern of breathing as co-factors. The following analyses refer to the Dunn post-hoc tests. We also agree that the clinical relevance of 41% protection compared to FFP3 with inhaled fraction <1% deserves better discussion. We restructured the discussion and results to better reflect this reasoning.

4. Page12, as a limitation, it is mentioned that the sensitivity of the technique for this study is 1%. Can we say this technique is valid or acceptable with this very low sensitivity rate? If I am not mistaken how sensitivity is used in this case, a 1% sensitivity rare makes your methodology unacceptable. Please revised or clarify how the sensitivity of the technique was measured and used in your methodology.

Reply: Thank you for raising this point. Sensitivity of 0.92% (rounded to 1%) is defined as the smallest amount of nebulized particles that we were able to detect, defined as percent of what was detected with zero distance between receiver and emitter and no mask. While we understand the concerns of the reviewer, we believe that 1% is sufficient for the purposes of this study. In fact, this resolution allowed us to describe a wide range of values (from above 60% to the sensitivity limit) according to the type of mask used. We agree that we could not claim that a device with <1% inhaled fraction would be as protective as a certified FFP3 mask, but this would be out of the scopes of the study, and no such claims are made throughout the manuscript. We better discuss this limitation in the revised manuscript, also de-emphasizing the translation from our in vitro findings and possible implications for masking policies, which was an error of over-interpretation we acknowledge and we worked on in this revision process.

Reply to Reviewer #2 comments

Reviewer #2: First of all it was interesting area of study and I would like to thank the authors. Then I have minor comments regarding your manuscript;

Generally it is better to see your English, some re-arrangements of your reporting and clarity of your statements for every reader.

Reply: We thank the reviewer for their positive and constructive comments. We edited extensively the manuscript including revision of wording and phrasing of certain parts.

Please change INTERPRETATION to Discussion in your abstract section.

Reply: Thank you. We changed accordingly.

In result section please start with Report from figure 3 shows…

Reply: Thank you. We changed accordingly.

The pattern of breathing had a negative impact on the filtering capacity: compared to normal breathing... Also, incorrect positioning of the mask decreased the mask’s performances. “Statements that are discussing should be taken to discussion section

Reply: We thank the reviewer for underlining this important point. We extensively revised the results section, removing these statements and presenting data in a more straightforward way.

°Significant difference compared to distance of 40 cm, § what are the signs at first and end of statements?

Reply: We apologize for this lack of clarity. As per PLOS ONE editorial guidelines, figure legends are in-line with the main text. Symbols should be interpreted as legends of the corresponding figure (Figure 3).

“The main findings of this study are that: 1) all investigated variables resulted significant (p < 0.001); 2) all tested masks reduced the inhaled fraction compared to distancing alone; 3) distance decreased the inhaled fraction; 4) the differences between devices were more pronounced at shorter distances; 5) polypneic breathing pattern led to a decrease in most masks’ efficacy; 6) findings clarified the limitations/advantages of using masks both for individual and other protection.” Take this statement to finding section.

Reply: We have extensively revised the discussion, and this introductory part was shortened and now serves as introduction of the discussion section.

[17], [18] your citation should better like this [17, 18].

Reply: Thank you. We changed accordingly.

Some of your statements are not easy to understand for all readers please revise all of your write up…“Distance alone markedly reduced the inhaled fraction also without masks, thus underlining how physical distancing policies intrinsically increase the protective effect of all other measures.”

“Worst experimental condition” what does this mean? Is there such experiment?

Reply: We apologize for the lack of clarity. All the sections mentioned by the reviewer have been extensively revised and clarified according to the reviewer’s suggestions. We hope you will find the revised manuscript clearer to follow.

---

## [Decision Letter · Decision Letter 1]

7 Apr 2021

Effects of distancing and pattern of breathing on the filtering capability of commercial and custom-made facial masks: 

an “in-vitro” study.

PONE-D-20-40104R1

Dear Dr. Alberti

We’re pleased to inform you that your manuscript has been judged scientifically suitable for publication and will be formally accepted for publication once it meets all outstanding technical requirements.

Kind regards,

Francesco Di Gennaro

Academic Editor

PLOS ONE

Additional Editor Comments (optional):

congratulations

Reviewers' comments:

Reviewer's Responses to Questions

**Comments to the Author**

1. If the authors have adequately addressed your comments raised in a previous round of review and you feel that this manuscript is now acceptable for publication, you may indicate that here to bypass the “Comments to the Author” section, enter your conflict of interest statement in the “Confidential to Editor” section, and submit your "Accept" recommendation.

Reviewer #1: All comments have been addressed

Reviewer #2: All comments have been addressed

2. Is the manuscript technically sound, and do the data support the conclusions?

Reviewer #1: (No Response)

Reviewer #2: Yes

3. Has the statistical analysis been performed appropriately and rigorously? 

Reviewer #1: (No Response)

Reviewer #2: Yes

4. Have the authors made all data underlying the findings in their manuscript fully available?

Reviewer #1: (No Response)

Reviewer #2: Yes

5. Is the manuscript presented in an intelligible fashion and written in standard English?

Reviewer #1: (No Response)

Reviewer #2: Yes

6. Review Comments to the Author

Reviewer #1: All my comments and concerns are addressed. I see the authors have done lots of improvement on the method, results, and discussion. Now, I feel it is scientifically acceptable for publication.

Reviewer #2: Thank you for considering my comments and improved your manuscript apropriately. I found Authors revision sound for publication.

7. PLOS authors have the option to publish the peer review history of their article (what does this mean?). If published, this will include your full peer review and any attached files.

Reviewer #1: **Yes: **Serebe Gebrie

Reviewer #2: **Yes: **Thomas Ayalew Abebe

---

## [Editor Report · Acceptance letter]

13 Apr 2021

PONE-D-20-40104R1 

Effects of distancing and pattern of breathing on the filtering capability of commercial and custom-made facial masks: an “in-vitro” study. 

Dear Dr. Alberti:

I'm pleased to inform you that your manuscript has been deemed suitable for publication in PLOS ONE. Congratulations! Your manuscript is now with our production department. 

Kind regards, 

on behalf of

Dr. Francesco Di Gennaro 

Academic Editor

PLOS ONE